# First Complete Mitochondrial Genome of Melyridae(Coleoptera, Cleroidea): Genome Description and Phylogenetic Implications

**DOI:** 10.3390/insects12020087

**Published:** 2021-01-20

**Authors:** Lilan Yuan, Xueying Ge, Guanglin Xie, Haoyu Liu, Yuxia Yang

**Affiliations:** 1The Key Laboratory of Zoological Systematics and Application, School of Life Science, Institute of Life Science and Green Development, Hebei University, Baoding 071002, China; 201972421@yangtzeu.edu.cn (L.Y.); gexueying2020@126.com (X.G.); 2College of Agriculture, Yangtze University, Jingzhou 434025, China; xieguanglin@yangtzeu.edu.cn

**Keywords:** Melyridae, mitochondrial genome, annotation, phylogenetic analysis

## Abstract

**Simple Summary:**

Melyridae, in a broad sense including Dasytinae and Malachiinae, is the largest family of Cleroidea distributed worldwide. However, the former two subfamilies are always treated as independent families by the European coleopterists. Opposite results have been produced by the two latest molecular phylogenetic works, so the development of reliable markers for reconstructing the phylogenetic relationships between the above taxa is of great importance. Here, we present the annotated complete mitogenome of *Cordylepherus* sp., which is the first complete mitogenome in Melyridae. The mitogenome of *Cordylepherus* sp. presents the typical organization of an insect mitochondrion. Comparisons of the newly generated mitogenome of *Cordylepherus* sp. to all available mitochondrial genomes of other Melyridae revealed no significant differences among them in terms of the length of each protein-coding gene, AT content of different genome regions, amino acid composition and relative synonymous codon usage. Phylogenetic analyses based on 13 protein-coding genes of mitogenomes show that the monophyly of Melyridae *sensu lato* is not supported, and Malachiinae and Dasytinae are suggested to be independent families, which are sister groups of Prionoceridae and Cleridae, respectively. Large-scale analyses with denser locus and taxon sampling are needed to confirm the present results.

**Abstract:**

To explore the characteristics of the mitogenome of Melyridae and reveal phylogenetic relationships, the mitogenome of *Cordylepherus* sp. was sequenced and annotated. This is the first time a complete mitochondrial genome has been generated in this family. Consistent with previous observations of Cleroidea species, the mitogenome of *Cordylepherus* sp. is highly conserved in gene size, organization and codon usage, and secondary structures of tRNAs. All protein-coding genes (PCGs) initiate with the standard start codon ATN, except ND1, which starts with TTG, and terminate with the complete stop codons of TAA and TAG, or incomplete forms, TA- and T-. Most tRNAs have the typical clover-leaf structure, except trnS1 (Ser, AGN), whose dihydrouridine (DHU) arm is reduced. In the A+T-rich region, three types of tandem repeat sequence units are found, including a 115 bp sequence tandemly repeated twice, a 16 bp sequence tandemly repeated three times with a partial third repeat and a 10 bp sequence tandemly repeated seven times. Phylogenetic analyses based on 13 protein-coding genes by both Bayesian inference (BI) and maximum likelihood (ML) methods suggest that Melyridae *sensu lato* is polyphyletic, and Dasytinae and Malchiinae are supported as independent families.

## 1. Introduction

The soft-winged flower beetles belong to the family Melyridae, which is the largest group of superfamily Cleroidea, with about 300 genera and 5500 species occurring in all major regions of the world [1]. The adults are commonly found on flowers where they feed on pollen or flower-visiting insects [2]. There has been inconsistency in its family definition in the history.

Most of the earlier authors considered the family Melyridae in the broad sense, including Prionocerinae, Rhadalinae, Malachiinae and Dasytinae [3,4,5]. Majer [6] split the former Melyridae *sensu lato* into several families, with Prionoceridae, Melyridae, Dasytidae and Malachiidae treated as independent families. This classification was followed by Mayor [7] in the latest catalogue of Palaearctic Coleoptera, also in agreement with Bocakova et al. [1] based on a four-gene (*18S*, *28S*, *16S* and *cox1*) phylogenetic study.

However, the modern coleopterological community [2,8,9] considers Prionoceridae and Melyridae as distinct families, while Dasytinae, Malachiinae and Rhadalinae are subfamilies of the latter. Most recently, Gimmel et al. [10] reconstructed the most comprehensive phylogeny of Cleroidea based on a four-gene dataset (*18S*, *28S*, *cox1* and *cytb*), and redefined Melyridae, including Dasytinae and Malachiinae but excluding Rhadalinae.

At the moment, the phylogenetic relationships of Melyridae, Dasytidae (Dasytinae) and Malachiidae (Malachiinae) remain controversial, and since the two most recent molecular phylogenetic studies, based on a few genes, produced conflicting results [2,10], the development of reliable markers for reconstructing the phylogenetic relationships for the above taxa is of great importance.

The mitochondrial genome has become a powerful tool for metazoan phylogenetic and evolutionary analysis [11,12,13,14,15,16] because of its small size, the presence of high copy numbers, strict orthologous genes, rare recombination and high evolutionary rate [12,17,18]. To date, only four mitochondrial genomes are available for Melyridae in GenBank, with two belonging to species for both Dasytinae and Malachiinae. However, none is a complete mitochondrial genome sequence. Deficiency of mitogenome data for Melyridae is an obstacle for a comprehensive phylogenetic analysis.

In the present study, the first complete mitochondrial genome for Melyridae, the complete mitogenome of *Cordylepherus* sp., was sequenced and analyzed. In addition, phylogenetic analyses based on two different methods were carried out to assess the phylogenetic position of *Cordylepherus* sp. in Cleroidea, and to contribute to further understanding the mitogenome evolution and phylogeny of the Cleroidea.

## 2. Materials and Methods

### 2.1. Sample Preparation and DNA Extraction

The material of *Cordylepherus* sp. was collected from Saihanba, Chengde, Hebei Province, China, on 29 May 2019. Specimens were preserved in 100% ethanol and deposited in the Museum of Hebei University, Baoding, China (MHBU, accession number CAN0146). Species identification was carried out one by one under a stereomicroscope. Total genomic DNAs were extracted using a DNeasy Blood & Tissue kit (QIAGEN, Beijing, China), according to the manufacturer’s instructions. DNAs were stored at −20 °C for long-term storage and further molecular analyses.

### 2.2. DNA Sequencing and Assembly

Whole mitochondrial genome sequencing was performed using an Illumina Novaseq 6000 platform (Illumina, Alameda, CA, USA) with 150 bp paired-end reads at Berry Genomics, Beijing, China. The sequence reads were first filtered following Zhou et al. [19] and then assembled using IDBA-UD [20] under similarity threshold 98% and k values of a minimum of 40 and a maximum of 160 bp. The gene *cox1* was amplified by polymerase chain reaction (PCR) using universal primers as “reference sequences” to acquire the best-fit. The PCR cycling conditions comprised a predenaturation at 94 °C for 5 min and 35 cycles of denaturation at 94 °C for 50 s, annealing at 48 °C for 45 s and elongation at 72 °C for 8 min at the end of all cycles. Geneious 2019.2 [21] software was used to manually map the clean readings on the obtained mitochondrial scaffolds to check the accuracy of the assembly.

### 2.3. Sequence Annotation and Analyses

Gene annotation was performed by Geneious 2019.2 software [21] and MITOS Web Server (http://mitos.bioinf.uni-leipzig.de/index.py) [22], with *Idgia oculata* as reference [23]. The tRNA Scan-SE server v. 1.21 [24] was used to re-identify the 22 tRNAs as well as to reconfirm their secondary structures, and the secondary structures were plotted with Adobe Photoshop 7.0. The mitogenome map was illustrated using CG View (http://stothard.afns.ualberta.ca/cgview_server) [25]. Tandem repeat elements in the A + T-rich region were identified using the Tandem Repeats Finder program (http://tandem.bu.edu/trf/trf.html) [26]. Nucleotide composition, base composition skewness, codon usage and relative synonymous codon usage (RSCU) of protein-coding genes were analyzed using PhyloSuitev 1.2.2 [27]. Strand asymmetry was calculated according to the formulas: GC skew = [G − C]/[G + C] and AT skew = [A − T]/[A + T] [28]. The nucleotide diversity (Pi) and nonsynonymous (Ka)/synonymous (Ks) mutation rate ratios were calculated by DnaSP v5.10.01 [29].

### 2.4. Phylogenetic Analysis

Mitochondrial genomes of 13 species from 7different families of Cleroidea were selected as ingroups, and mitochondrial genomes of 2 Cucujoidea species were chosen as outgroups (details in Table 1). All mitochondrial genomes (except the one sequenced in this study) were obtained from GenBank (accession numbers given in Table 1). Data standardization and information extraction were performed by PhyloSuite v 1.2.2 [27]. The nucleotide sequences of the 13 protein-coding genes were analyzed. The protein-coding genes were aligned using the MAFFT v 7.313 plugins [30], optimized using MACES in PhyloSuite v 1.2.2 [27]. Intergenic gaps and ambiguous sites were removed using Gblocks v 0.91b [31], then all protein-coding genes were concatenated in PhyloSuitev 1.2.2 [27]. The optimal partition schemes and the best-fit replacement models were selected by Model Finder [32], and the results are presented in Appendix A. The “greedy” algorithm with branch lengths estimated as “linked” and the Bayesian information criterion were used.

Phylogenetic trees were constructed based on both maximum likelihood (ML) and Bayesian inference (BI). The ML phylogenies were inferred using IQ-TREE.v.1.6.8 [40] and BI phylogenies using MrBayes 3.2.6 [41], respectively. The ML phylogenetic analyses were performed with the ultrafast bootstrap (UFboot) algorithm with 1000 replicates. In BI phylogenetic analyses, 5 × 10^6^ Markov Chain Monte Carlo (MCMC) generations after reaching stationarity (average standard deviation of split frequencies < 0.01) were used as the default settings, with estimated sample size >200, and potential scale reduction factor ≈1 [41]. Interactive Tree of Life (iTOL, http://itol.embl.de) was used to display, annotate and manage the phylogenetic tree.

## 3. Results and Discussion

### 3.1. Mitogenome Organization and Base Composition

Sequence generated in this study is deposited in GenBank with accession number (MW365444). The complete mitogenome sequence of *Cordylepherus* sp. is 15,824 bp in length. It is a circular, double-stranded ring which includes 13 protein-coding genes (PCGs), 22 tRNA genes, 2 rRNA genes and an A+T-rich region (control region) (Figure 1A). The gene organization is the same as the hypothetical ancestral insect mitogenome as direction and order [42]: 15 genes (8 tRNAs, 4 PCGs and 2 rRNAs) are encoded on the minority strand (N-strand), and the others (14 tRNAs and 9 PCGs) are transcribed from the majority strand (J-strand) (Appendix A). The mitogenome contains 7 overlapped genes (13 bp in total), and the longest overlap is 4 bp. Most of the gene overlaps occur in tRNA genes, which is possibly due to the lower evolutionary constraints of these genes [37]. Intergenic spacers in the *Cordylepherus* sp. mitogenome have 12 regions ranging from 1 to 34 bp (a total of 98bp), with the longest region detected between *tRNA-Tyr* and *cox1*.

The base composition of *Cordylepherus* sp. mitogenome is A (41.90%), T (39.10%), C (11.20%) and G (7.80%). The A/T nucleotide composition is 81.00%, thus exhibiting a high A/T bias, as other insects [13,14,15,16,33,35]. The AT-skew of the full mitogenome is positive (0.03), while the GC skew is negative (−0.18). This indicates that the content of bases G is higher than that of C, and A is higher than T in the whole (Appendix A).

### 3.2. Protein-CodingGenes

The total length of the 13 PCGs in *Cordylepherus* sp. is 10,996 bp, approximately accounting for 69.5% of the whole mitogenome (Appendix A). Nine of the 13 PCGs are coded on the J-strand (*cox1*, *cox2*, *cox3*, *cytb*, *nad2*, *nad3*, *nad6*, *atp6*, *atp8*), and the other four (*nad1*, *nad4*, *nad4l*, *nad5*) are located on N-strand (Figure 1, Appendix A). The A/T nucleotide composition is 79.70%, also exhibiting a highly A/T bias (Appendix A). Most of the 13 PCGs initiate with the standard start codon ATN, common among metazoan [43], except *nad1*, which starts with TTG. There are four types of stop codons, TAA (*nad6*, *cox2*, *atp6*, *atp8*, *cox3*, *nad4l*) and TAG (*nad1*), as well as TA- (*nad4*) and T- (*nad2*, *cox1*, *nad3*, *nad5*, *cytb*). It is common that incomplete stop codons occur in insects; this is believed to be completed by the rough polyadenylation processes and polycistronic transcription cleavage, and it may emerge as functional stop codons or serve to minimize gene overlap and spacer [44].

To characterize codon frequencies across *Cordylepherus* sp., relative synonymous codon usage (RSCU) was calculated and drawn, as shown in the Figure 2 and Appendix A. In total, it contains 3656 codons excluding stop codons, and the most frequently used codons are UUA (502), AUU (405), UUU (343) and AUA (265). Accordingly, Leu, Ile, Phe and Met are the most frequently used amino acids, accounting for 16.11%, 11.57%, 9.98% and 7.66% of the total amino acids, respectively. In these most frequently used codons, A and U are the components which contribute to the high A + T bias of the full mitogenome. Additionally, the RSCU values of NNU and NNA are greater than 1, indicating that the third position of the codon is rich in A and T. Besides, PCGs shows positive AT skew (0.04) and negative GC skew (−0.16) (Appendix A).

In order to illustrate differences between Malachiinae and Dasytinae, we compared the length of the 13 PCGs (Appendix A), AT content (Appendix A), RSCU and amino acid composition (Appendix A) in different regions, as shown in the Figure 2 and Figure 3. The length of *atp6*, *cox2*, *cytb*, *nad2*, *nad4*, *nad4l* and *nad5* is higher in Dasytinae than Malachinae, but *nad6* is the opposite (Figure 3A). The AT contents of mitogenomes are lower in Dasytinae than Malachiinae in most regions, except for the full genome, 3rd codon position, *nad2*, *nad4*, *nad5* and *cytb* (Figure 3C). Among the RSCU, the CUG codon used for Leu is absent in Dasytinae (Figure 2A). No differences were observed between them in the amino acid composition and proportion (Figure 2B).

The nucleotide diversity of the Melyridae is shown in Figure 3B; it is highly variable among the 13 PCGs, with values ranging from 0.156 (*cox1*) to 0.470 (*nad1*). The gene *nad1* (Pi = 0.470) has the most diverse nucleotide among all PCGs, followed by *nad2* (Pi = 0.269), *atp8* (Pi = 0.268), *nad6* (Pi = 0.256) and *nad4* (Pi = 0.212). In contrast, *cox1* (Pi = 0.156), *cox2* (Pi = 0.162) and *cox3* (Pi = 0.163) have relatively low values of nucleotide diversity and are the most conserved genes.

Average non-synonymous (Ka)/synonymous (Ks) substitution rate ratios can be used to estimate the evolutionary rate [45]. The Ka/Ks ratios of *nad2* (1.017) and *nad4* (1.339) genes are greater than one, indicating that they are under positive selection [46], while the other genes are under purifying selection, with ratios ranging from 0.100 to 0.859 (all less than 1) [46]. The genes *nad2* and *nad4* have relatively high evolutionary rates, while *cox1* (0.100) and *cox2* (0.169) have comparatively low Ka/Ks ratios (Figure 3B). Like other insects [47,48], the genes with lower evolutionary rate could be used as barcodes for inferring the phylogenetic relationships, such as *cox1* and *cox2* in Melyridae, while those evolving more quickly are more suitable for species identification, especially for *nad2* and *nad4* in Melyridae.

### 3.3. Transfer and Ribosomal RNA Genes

As other insects [14,15,16,33,35], the *Cordylepherus* sp. mitogenome contains 22 tRNAs genes, which range from 63 to 71 bp in length; the total length of tRNAs is 1149 bp (Appendix A). Most tRNAs genes could be folded into the typical clover-leaf secondary structure, while *trnS1* (AGN) lacked a dihydrouridine (DHU) arm (Figure 4), as observed in many other insect mitogenomes [12,43]. In general, the secondary structure is conserved in the length of the acceptor and anticodon arms (7 bp in the former and 5 bp in the latter), while variable in that of the DHU and TΨC arms [49,50,51]. In the secondary structures of all tRNAs of *Cordylepherus* sp., 3 or 4 base pairs in the DHU arms, and 3, 4 or 5 base pairs in the TΨC arms are found. Except the classic base pairs (A-U and C-G), 15 noncanonical base pairings (G–U and A–C) and 5 other mismatched base pairs (U-U and C-U) are found in the arms. The AT content is 81.70%, with a positive AT skew (0.07) and negative GC skew (−0.16) (Appendix A).

There are two rRNAs, a 1258-bp *16S*rRNA (*rrnL*) and an 815-bp *12S*rRNA (*rrnS*) in *Cordylepherus* sp. (Appendix A). It is hard to determine the boundaries of the rRNA genes, since that they do not have functional annotation features as PCGs [52,53]. Thus, the boundaries of flanking genes are decided by assuming no overlapping or gaps between adjacent genes, like that in the inferred insect mitogenome pattern. The *16S* rRNA subunit is located between *trnL1* and *trnV*; the *12S* rRNA genes are located between *trnV* and the A+T-rich region. The AT content is 84.50%, with positive AT skew (0.02) and negative GC skew (−0.32) (Appendix A).

### 3.4. A + T-RichRegion

The mitochondrial A+T-rich region (or control region, CR) acts on the initiation and regulation of insect replication and transcription [54,55]. This non-coding region is located between *rrnS* and *trnI* in the mitogenome of *Cordylepherus* sp., with a total length of 1194 bp (Appendix A). The AT content is 90.00%, with both negative AT skew (−0.02) and GC skew (−0.09) (Appendix A). Three tandem repeat sequence units are present; their positions and length are shown in the Figure 1B. They area 115 bp-sequence tandemly repeated twice, a 16 bp-sequence tandemly repeated three times with a partial third repeat and a 10 bp-sequence tandemly repeated seven times, respectively.

### 3.5. Phylogenetic Analysis

Similar topologies are yielded from ML (Figure 5A) and BI (Figure 5B) analyses with high nodal support values. The monophyly of Cleroidea is highly supported (PP = 1, BS = 100), with Byturidae included in it [56]. Within Cleroidea, it is divided into three branches. One clade consists of Malachiinae and Prionoceridae, with a high statistical support value (PP = 1, BS = 100). The other two clades (Chaetosomatidae (Dasytinae, Cleridae)) and (Trogossitidae, Phloiophilidae and Byturidae) seem more closely related in the tree and supported with a relatively high value (PP = 1, BS = 92). However, within each of the latter two clades, the interrelationships among the included taxa are not strongly supported in the BI analysis, except the sister relationship between Dasytinae and Cleridae (PP = 1, BS = 100).

This result does not recover Melyridae *sensu lato* as monophyletic. Malachiinae, with three species included in the analyses, is resolved as monophyletic, and sister to Prionoceridae. Dasytinae, with two sampled species, formed a monophyletic clade sister to Cleridae. Both clades are strongly supported (PP = 1, BS = 100). Our analyses suggest that Malachiinae and Dasytinae should be treated as independent families, which agrees with the views of Majer [6] and Bocakova et al. [1], but is incongruent with the others [8,9,10]. It is necessary to further examine the phylogenetic relationships of Melyridae family groups or Cleroidea, if more molecular data (more representative species and more molecular markers) are available.

## 4. Conclusions

Consistent with previous observations of other Cleroidea species, the mitogenome sequences of *Cordylepherus* sp., which is the first complete mitochondrial genome of Melyridae, are highly conserved in gene size and organization, highly A + T biased base composition, codon usage of protein-coding genes and secondary structures of tRNAs. This provides the basic information to perform comparative analyses and further discussion of the mitogenome evolution of Cleroidea.

Phylogenetic analyses support Malachiinae and Dasytinaeas as independent families, instead of subfamilies of Melyridae *sensu lato*, which is suggested to be a polyphyletic group. Larger scale studies with more locus and taxon sampling are still needed to reconstruct more comprehensive phylogenies in order to achieve a better resolution of the relationships of Melyridae family groups or Cleroidea.

## Figures and Tables

**Figure 1 insects-12-00087-f001:**
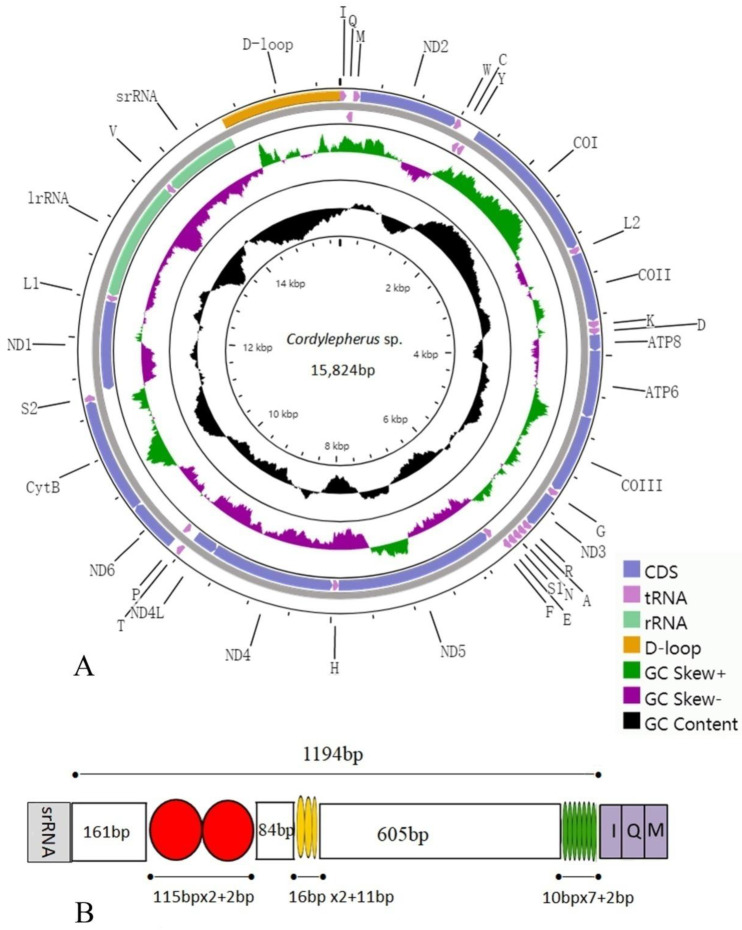
(**A**) Mitochondrial genome map of *Cordylepherus* sp.; (**B**) organization of the A + T-rich regions in the mitochondrial genomes of *Cordylepherus* sp.

**Figure 2 insects-12-00087-f002:**
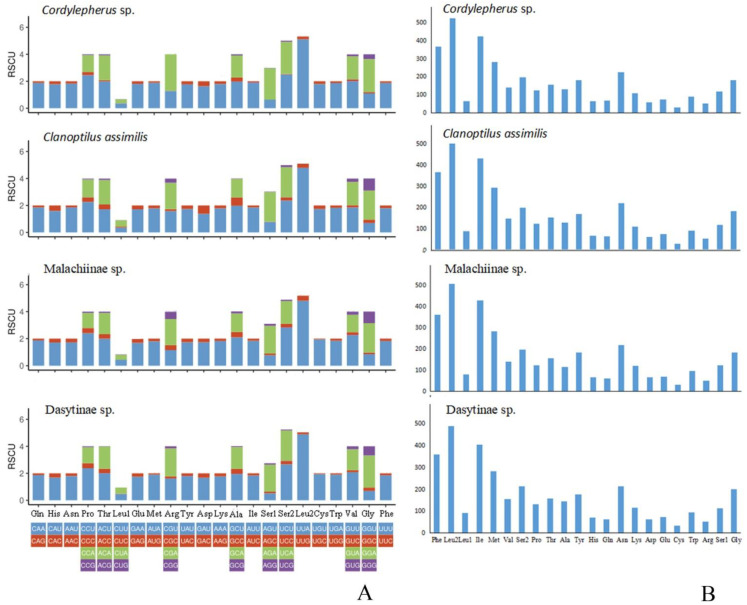
(**A**) Relative synonymous codon usage (RSCU) in the Melyridae species’ mitogenomes; (**B**) amino acid composition in the Melyridae species’ mitogenomes.

**Figure 3 insects-12-00087-f003:**
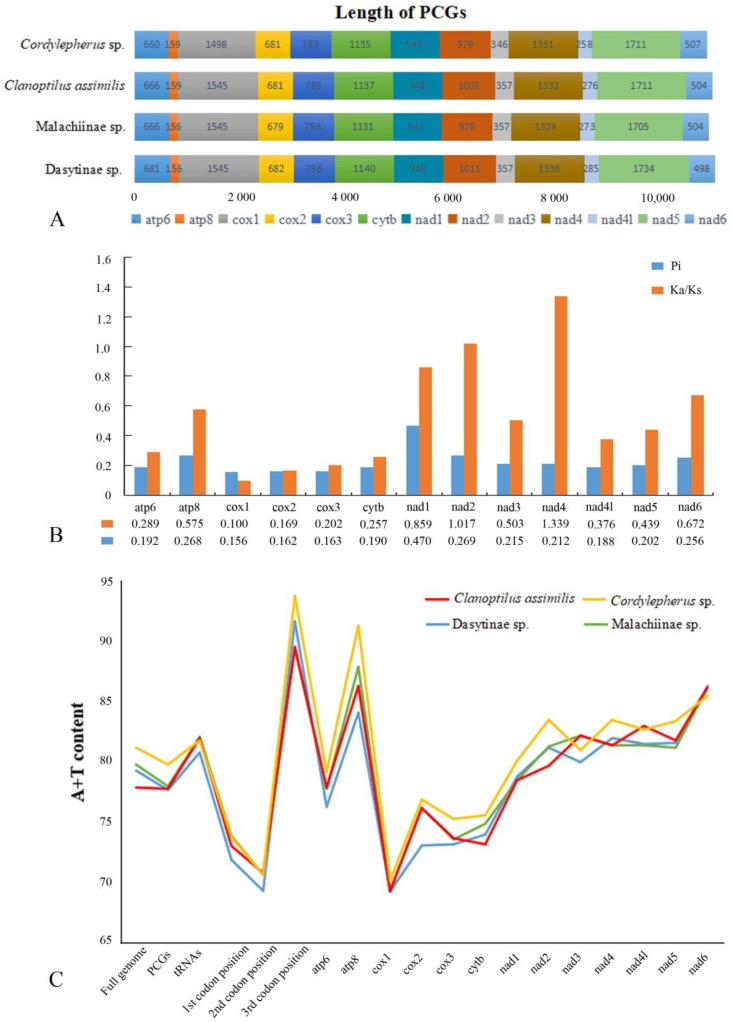
(**A**) Comparison of 13 PCGs in lengths of Melyridae species; (**B**) nucleotide diversity (Pi) and nonsynonymous (Ka) to synonymous (Ks) substitution rate ratios of 13 PCGs of Melyridae species (the Pi and Ka/Ks values of each PCGs shown under the gene name); (**C**) comparison of AT contents in different regions of mitogenomes of Melyridae species.

**Figure 4 insects-12-00087-f004:**
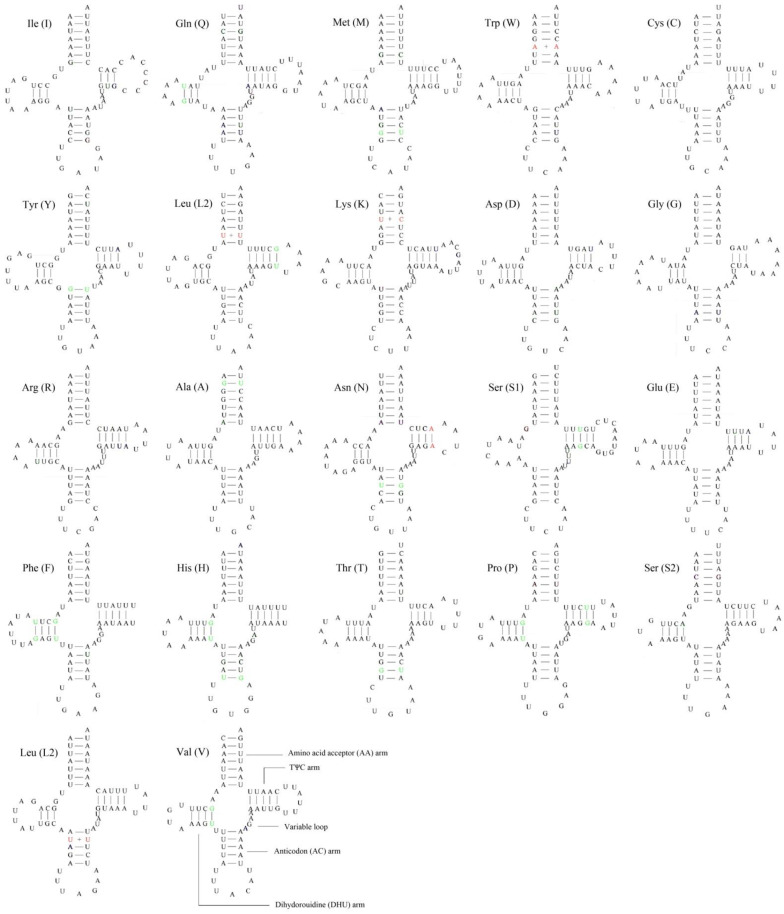
Putative secondary structures of tRNAs from *Cordylepherus* sp. mitogenome (noncanonical base pairings in green; mismatched base pairs in red).

**Figure 5 insects-12-00087-f005:**
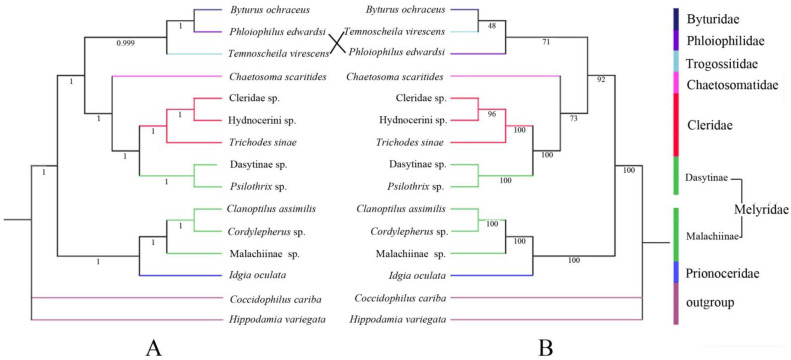
Phylogenetic tree of Cleroidea produced from maximum likelihood (ML) (**A**) and Bayesian inference (BI) (**B**) analyses based on 13 PCGs (posterior probabilities (PP) and bootstrap (BS) values indicated in each clade).

**Table 1 insects-12-00087-t001:** Summary of the representative species and their mitogenome information in this study.

Superfamily	Family/Subfamily	Species	GenBankNo.	References
Cleroidea	Phloiophilidae	*Phloiophilus edwardsi*	JX412815.1	Unpublished
(Ingroup)	Melyridae/	Malachiinae sp.	JX412799.1	Unpublished
	Malachiinae	*Clanoptilus assimilis*	JX412833.1	Unpublished
		*Cordylepherus* sp.	MW365444	This study
	Melyridae/	*Psilothrix* sp.	JX412801.1	[33]
	Dasytinae	Dasytinae sp.	JX412765.1	Unpublished
	Trogossitidae	*Temnoscheila virescens*	JX412752.1	Unpublished
	Prionoceridae	*Idgia oculata*	NC_044896.1	[23]
	Byturidae	*Byturus ochraceus*	NC_036267.1	[34]
	Cleridae	Hydnocerini sp.	KX035157.1	Unpublished
		*Trichodes sinae*	NC_033340.1	[35]
		Cleridae sp.	MH789728.1	[36]
	Chaetosomatidae	*Chaetosomas caritides*	NC_011324.1	[37]
Cucujoidea	Coccinellidae	*Coccidophilus cariba*	MN447521.1	[38]
(Outgroup)		*Hippodamia variegata*	MK334129.1	[39]

## Data Availability

The sequence generated in this study is deposited in GenBank with accession number (MW365444).

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
