# Peer review of "First Complete Mitochondrial Genome of Melyridae(Coleoptera, Cleroidea): Genome Description and Phylogenetic Implications"

_insects, 2021, doi:10.3390/insects12020087_

Round 1

Reviewer 1 Report

The complete mitochondrial genome of Cordylepherus sp was determined in this study. The gene content, gene order, strand asymmetry, base composition, rRNA and tRNA secondary structure and phylogenetic analysis were described.

Every new genome is always welcome and appreciated by scientific community, as it will improve comprehension of phylogenetic relationships among taxa. Under this perspective, this work matches this aim, but there are some points that must be improved.

In material and methods section the strategy used for mitochondrial genome sequencing is not well explained: from what I understood it is a quite novel approach of whole mitochondrial sequencing, without the use of PCR. It could be worthwhile to highlight it. If I am wrong, then definitely it has to be written better.

Moreover, I would suggest to add nuclear sequences to your data set for phylogenetic reconstruction. Most scientists would not accept a phylogenetic tree based on mitochondrial genes only, as mitochondria are considered as one evolutionary unit, even if you are using many genes, expecially to solve relations at family level. What happen to the polyphyletic clade, if you add ribosomial genes?

There is no mention to morphological data and how they would fit your phylogenetic results: integrative approaches might be useful to solve phylogenetic question of Cleroidea families.

Results and discussion section in highly unbalanced, with very little discussion. The authors should make more comparisons with the previously sequenced mito-genomes and the phylogenetic section is poor. 

See comments on References, some are redundant.

English should be carefully checked. There are many mistakes.

Author Response

For the title, I would not like to make any change. Although Malachiinae and Dasytinae are supported as independant families based on our analyses, no taxonomical changes were made. The phylogeny of Malachiinae and Dasytinae with respect to the Melyridae needs to be further elucidated with additional samples, as has been suggested in the text.

In the material and methods part, some details about PCR is added.

Although the nuclear sequences is important for phylogenetic reconstruction, it is beyond the aim of this study. So is it for the morphological data.

In the results and discussion section, some comparisons with the previously sequenced mito-genomes were added, and the phylogenetic section was  discussed further. 

I agree with you to delete the four highlightened references.

The whole text was checked carefully, and indeed many mistakes were corrected.

In the text, I agreed with you for the corrections or suggestions, and accordingly made changes. Please see the details in the attached revised MS.

Reviewer 2 Report

The study of Yuan et al. represents a good attempt to a classical molecular study.They aim to describe the first complete mitochondrial genome of Melyridae, and test the phylogenetic relationships within this family.
The study is well designed, but I have specific comments:
l 68: specify what “mt” means,
l 97: please specify your reference genome for the gene annotation
l 109: could you specify your molecular sampling (length, number of genes…)? it is not clear how many sequences you have for each taxon
Table 1: highlight your study (in bold f.e.)
Figure 1: add a picture of your species

Author Response

l68: “mt genome ” changed into “mitogenome”;

l97: in the Material and methods part, a sentence is added "Gene annotation was done by Geneious 2019.2 and Idgia oculata as reference."

l109: the sequences used in the phylogenetic study are indicated in detail in the Table 1. I do not think it is necessary to repeat it in the text.

Table 1: Yes, the sequence in this study is in bold now.

Figure 1: It is hard to get a good habitus photo for the specimen preserved in the alcohol for molecualr study. I think the quality of the plate would be reduced if it was added to the figure 1. What's worse, now it is impossible to get a better photo for the specimen because of its damagement happened in the experiment.  I am sorry for this.

Reviewer 3 Report

Dear authors and editors,

I think this study is been reasonably designed and properly performed. Some minor modifications are needed. I have put my comments in the attached file. 

Author Response

Thanks for your corrections, and accordingly we made some changes in the text. Please find the details in the revised version.

Reviewer 4 Report

This manuscript describes the mitochondrial genome of Melyridae and use 13 protein coding genes to investigate the phylogeny of this clade. The manuscript is clearly written and the methods contain sufficient details for the reader. I have very few comments.

It would be worthwhile to add a nucleotide diversity analysis with non-synonymous (Ka) to synonymous (Ks) substitution rates for the 13 PCGs to the manuscript. Please see Fig 5 in Zhou et al. 2020 for an example and methods (https://www.mdpi.com/2075-4450/11/4/232/htm)

The phylogeny of Malachiinae and Dasytinae with respect to the Melyridae needs to be further elucidated with additional samples, as has been suggested by the authors.

Typographical errors

Ln 72 tree-construction

Ln 251 Consistent

Ln 255 perform comparative analyses and further discussion of

Author Response

Following your suggestion, we added a nucleotide diversity analysis with non-synonymous (Ka) to synonymous (Ks) substitution rates for the 13 PCGs. Accordingly, some context were supplemented in the material and methods and result and discussion sections, also some references were added, as well as an illustration was presented.

L72 The word "tree-construction" was changed into a different word.

L251 I agreed with you to change "Consist" into "Consistent".

Ln 255 I agreed with you to change the sentence into "perform comparative analyses and further discussion of".

Please find the details in the revisied version.
